# The Processing and Electrical Properties of Isotactic Polypropylene/Copper Nanowire Composites

**DOI:** 10.3390/polym14163369

**Published:** 2022-08-18

**Authors:** Po-Wen Lu, Chonlachat Jaihao, Li-Chern Pan, Po-Wei Tsai, Ching-Shuan Huang, Agnese Brangule, Aleksej Zarkov, Aivaras Kareiva, Hsin-Ta Wang, Jen-Chang Yang

**Affiliations:** 1Division of Gastroenterology and Hepatology, Department of Internal Medicine, Shuang Ho Hospital, Taipei Medical University, Zhongzheng Rd., Zhonghe, Taipei 23561, Taiwan or; 2Graduate Institute of Biomedical Materials and Tissue Engineering, College of Biomedical Engineering, Taipei Medical University, Taipei 11052, Taiwan; 3Graduate Institute of Biomedical Optomechatronics, Taipei Medical University, Taipei 110-31, Taiwan; 4Institute of Organic & Polymeric Materials, National Taipei University of Technology, Taipei 106, Taiwan; 5School of Dentistry, College of Oral Medicine, Taipei Medical University, Taipei 110-31, Taiwan; 6Department of Pharmaceutical Chemistry, Riga Stradins University, LV-1007 Riga, Latvia; 7Institute of Chemistry, Vilnius University, Naugarduko 24, LT-03225 Vilnius, Lithuania; 8International Ph.D. Program in Biomedical Engineering, College of Biomedical Engineering, Taipei Medical University, Taipei 110-31, Taiwan; 9Research Center of Biomedical Device, Taipei Medical University, Taipei 110-31, Taiwan; 10Research Center of Digital Oral Science and Technology, Taipei Medical University, Taipei 110-31, Taiwan; 11Graduate Institute of Nanomedicine and Medical Engineering, College of Biomedical Engineering, Taipei Medical University, 250 Wu-Hsing Street, Taipei 110-31, Taiwan

**Keywords:** isotactic polypropylene, copper nanowires, percolation threshold, power law

## Abstract

Polypropylene (PP), a promising engineering thermoplastic, possesses the advantages of light weight, chemical resistance, and flexible processability, yet preserving insulative properties. For the rising demand for cost-effective electronic devices and system hardware protections, these applications require the proper conductive properties of PP, which can be easily modified. This study investigates the thermal and electrical properties of isotactic polypropylene/copper nanowires (i-PP/CuNWs). The CuNWs were harvested by chemical reduction of CuCl_2_ using a reducing agent of glucose, capping agent of hexadecylamine (HDA), and surfactant of PEG-7 glyceryl cocoate. Their morphology, light absorbance, and solution homogeneity were investigated by SEM, UV-visible spectrophotometry, and optical microscopy. The averaged diameters and the length of the CuNWs were 66.4 ± 16.1 nm and 32.4 ± 11.8 µm, respectively. The estimated aspect ratio (L/D, length-to-diameter) was 488 ± 215 which can be recognized as 1-D nanomaterials. Conductive i-PP/CuNWs composites were prepared by solution blending using p-xylene, then melt blending. The thermal analysis and morphology of CuNWs were characterized by DSC, polarized optical microscopy (POM), and SEM, respectively. The melting temperature decreased, but the crystallization temperature increasing of i-PP/CuNWs composites were observed when increasing the content of CuNWs by the melt blending process. The WAXD data reveal the coexistence of Cu_2_O and Cu in melt-blended i-PP/CuNWs composites. The fit of the electrical volume resistivity (ρ) with the modified power law equation: ρ = ρ_o_ (V − Vc)^−t^ based on the percolation theory was used to find the percolation concentration. A low percolation threshold value of 0.237 vol% and high critical exponent t of 2.96 for i-PP/CuNWs composites were obtained. The volume resistivity for i-PP/CuNWs composite was 1.57 × 10^7^ Ω-cm at 1 vol% of CuNWs as a potential candidate for future conductive materials.

## 1. Introduction

Polypropylene (PP), a versatile engineering plastic, has the advantages of excellent chemical resistance, mechanical properties, lightweight, flexible processability, and recycling characteristics that make it optimum for many long-life industrial applications [1,2,3]. Among various PP, semi-crystalline isotactic polypropylene (i-PP) showed appropriate optical elements upon recycling [4], but bears volume resistivity as high as 10^16^ Ω-cm classified as an electrically insulating material [5]. 

Static electricity is known as rested electricity due to an imbalance of positive and negative electric charges on the surface or within materials [6]. The induced electrostatic voltages can possibly reach 30,000–40,000 volts simply by a triboelectric effect such as rubbing or separating of materials; however, some micro-electronic parts can be damaged by unexpected electrostatic discharge (ESD) as low as 20 volts [7]. The specification for an ESD material needs to retain a surface resistivity between 10^5^–10^12^ Ω/sq based on the standard 541 issued by the electronic industries association (EIA) [8].

Polypropylene can be tailored to conductive by loading electrically conductive fillers or additives to extend its applications for electrostatic discharge (ESD) protection [9]. Various conductive nanofillers have been reviewed for their potential use in conductive polymer preparation depending on their percolation threshold, maximum conductivity/resistivity, and processing details. Fillers with high aspect ratios (L/D, length-to-diameter) tend to form a conductive network under low loading dosages [10]. Commonly used fibrous fillers include carbon, metal, and metal-coated fibers. The class of metal-based conductive nanofillers has received greater attention from the research community because of their superior multifunctional properties compared to conventional fillers. 

In recent years, the synthesis of metal nanowires has gained much attention because of their outstanding conductivity properties and their metal nanowires loaded polymer nanocomposites under a percolation threshold as low as 0.25 and 0.75 vol % [11]. Like gold or silver, copper exhibits excellent thermal and electrical conductivity but high cost-effective. These properties have enabled solution-processed copper nanowires (CuNWs) in several promising industry applications such as flexible electronics, transparent conductors, optoelectronic devices, and chemical and biological sensing devices [12]. Many chemical synthetic routes were reported including reduction by chemicals such as NaBH_4_ [13] or C_6_H_12_O_6_ [14], electrochemistry [15], sonochemistry [16], and X-ray irradiation [17]. However, the processing difficulties and poor dispersion of CuNWs limit their usage in the fabrication of conductive polymer composites. In this study, we report surfactant-based synthesis routes with its processing map for CuNWs preparation and investigate the electrical properties and percolation threshold for isotactic polypropylene/CuNWs composites. 

## 2. Materials and Methods

### 2.1. Materials

Copper (II) chloride dehydrate (CuCl_2_·2H_2_O) was purchased from J.T.Baker, Phillipsburgc, NJ, USA. Glucose (Dextrose monohydrate, C_6_H_12_O_6_·H_2_O) and hexadecyl amine (HDA, C_16_H_35_N) were obtained from Sigma-Aldrich, St. Louis, MO, USA. The p-Xylene (99%) was purchased from Acros, St. Louis, MO, USA, and PEG-7 Glyceryl Cocoate was provided from KOLB, Hedingen, Switzerland, while the ethanol (C_2_H_5_OH, 95%) was from Echo Chemical Co., Miaoli, Taiwan. All chemicals were used as received without further purification. The pellet form of isotactic polypropylene, i-PP (PC6501), was provided by Lee Chang Yung Chemical Industry Co., Taipei, Taiwan. The Mv and density were 280,000 g/mole and 0.90 g/cm^3^, respectively.

### 2.2. Copper Nanowires Preparation

In one of the examples of CuNWs synthesis shown in Figure 1, 21 mg CuCl_2_ and 120 mg HDA were dissolved in 10 mL of deionized (DI) water under vigorously stirred at 55 °C overnight to obtain a uniform blue emulsion. Then, 50 mg glucose was used as a reducing agent, added to this precursor emulsion, and kept in a 100 °C dry bath (VH-01, Violet Bio Sciences, Tokyo, Japan) for 6 h without stirring. Afterward, the dark red CuNWs suspension was washed with DI water thoroughly 3 times with a centrifugation-redispersion cycle. Then, the precipitate was redispersed in 90 mL of deionized water. To separate CuNWs from the residual HAD, we added 10 mL of p-xylene into the CuNWs/DI water suspension and vortexed for 30 s, then collected them using a micro refrigerated centrifuge (model 3740, KUBOTA, Naniwa-Ku, Osaka, Japan). All harvested CuNWs were carefully managed and washed with ethanol and p-xylene 3 times. The CuNWs/xylene solution was stored under N_2_ atmosphere before usage. 

### 2.3. Preparation of i-PP/CuNWs Composites

To prepare the i-PP/CuNWs composites, we first loaded i-PP pellets under the ratio of 45 mg i-PP/2 mL p-xylene into a container and kept them in a dry bath of 120 °C until completely dissolved. Each pre-determined amount of CuNWs was added into the 10 mL p-xylene solution under N_2_ bubbling to prevent CuNWs oxidation in an ultrasonic cleaner (LEO-803, Leo Ultrasonic Co., New Taipei City, Taiwan) till they were thoroughly dispersed. The CuNWs/p-xylene solution was then carefully transferred and mixed with i-PP/xylene solution using a digital hot plate stirrer (PC 420, Corning, New York, NY, USA). After that, 20 mL of ethanol was added to precipitate the i-PP/CuNWs from the solution. The obtained i-PP/CuNWs were collected through vacuum filtration and dried in a vacuum oven for 3 h at 40 °C. The i-PP/CuNWs were stacked and hot-pressed at 190 °C under an N_2_ atmosphere to form a film and stored in a dry box before usage. 

### 2.4. Characterization

The UV-visible spectroscopy (JASCO V-770 Spectrophotometer, Tokyo, Japan) was then performed to confirm CuNWs formation. In addition, the influence of capping agent HAD contents on the morphology of CuNWs was investigated. 

The microstructural characterization for the morphology of as-synthesized CuNWs and i-PP/CuNWs was carried out for each sputter-coated sample using an ion-sputter (S-3000H, Hitachi, Tokyo, Japan), then examined by scanning electron microscopy (SEM, SU3500, Hitachi Ltd., Tokyo, Japan) at 15 kV at a magnification of 2000×. The mean diameter of CuNWs was calculated from the average 20 measurements of each specimen. The surfactant of PEG-7 content on the dispersion behavior of CuNWs suspension was conducted.

X-ray diffraction (XRD, D/MAX-RC, Rigaku, Japan) with a Ni filter and CuKα radiation (λ = 0.154 nm) at 30 kV and 20mA. Measurements were conducted in a continuous scan mode with a scanning rate of 10°/min and 2θ from 10° to 80°. 

The melting and crystallization behaviors of i-PP/CuNWs composites were probed with a differential scanning calorimeter (DSC, TA Q100, TA Instrument, New Castle, DE, USA). About 5 mg of the hot-pressed i-PP/CuNWs composite was loaded in aluminum DSC pans, hermetically sealed with a crimping press, and placed in a DSC oven. The heating and cooling run between 40 °C to 200 °C at a rate of 10 °C/min under a nitrogen flow rate of 50 mL/min.

The dispersion state of CuNWs in i-PP was examined using a polarized optical microscope (OM, BX51, Olympus, Tokyo, Japan) under a bright field. The crystallization of i-PP/ CuNWs composites from the temperature of 200 °C cooling to ambient temperature was examined using polarized optical microscopy (POM) (COOLPIX995, Nikon, Tokyo, Japan). It was equipped with a heating and freezing stage (Linkam THMS600, Linkam Scientific Instruments Ltd., Tadworth Surrey, UK).

The volume resistivity was measured using a four-point sheet resistance probe by electrometer/high resistance meter (Model 6517B, Keithley, Melrose, MA, USA) under an ambient environment. The test samples possess a thickness of 0.15 mm and a size of 3 × 3 cm. 

### 2.5. Data Analysis

The one-way analysis of variance (ANOVA) with post-hoc Tukey HSD (“honestly significant difference”) test was carried out to evaluate the statistical significance of the measured results using an online calculator (Interactive Statistics—One-way ANOVA from Summary Data (https://astatsa.com/OneWay_Anova_with_TukeyHSD/ (accessed on 28 July 2022)). The results of *p* < 0.05 were considered statistically significant. 

## 3. Results and Discussion

### 3.1. The UV-Vis Absorption Spectra for CuNWs Solutions

In a typical synthesis procedure, CuCl_2_, glucose, and HDA were mixed and heated in a solution for 6 h. The solution color gradually turned from blue to red-brown, implying the formation of Cu nanostructures. The UV-Vis absorption spectra were recorded to characterize Cu nanoparticle suspension due to the surface plasmon resonance bands at different frequencies from different morphologies. Figure 2a shows the UV-visible spectra recorded from the CuCl_2_ and the harvested Cu nanostructure suspension solution. The existence of a localized surface plasmon resonance peak at 570 nm is the well-reported signature of CuNWs formation [18]. When changing the HDA content from 90 mg to 150 mg, the surface plasmon resonance peaks for CuNWs variated in the range of 570–590 nm (Figure 2b). It is well-recognized that fibrous nanostructures exhibit both transverse and longitudinal plasmon resonance peaks due to the light-induced collective conduction electron oscillations. M. Luo et al. [19] reported that only the transverse mode surface plasmon resonance peak for CuNWs at 560 nm was observed. Still, the extinction of longitudinal elevation might be attributed to the shifting to the infrared region owing to the extremely large aspect ratios of the CuNWs. Kumar D.V.R. et al. reported the UV-Vis maximum absorption peak dependence of CuNWs suspensions on the various alkyl amines and the peak value of 568 nm for HDA [20]. 

The dosage of HDA content was also found to be essential in controlling the final morphology of Cu nanocrystals. Figure 3 shows the SEM images of synthesized Cu nanostructure mixture of particles, cubes, or wires depending on the HDA dosage range from 90 mg to 180 mg. Among them, the SEM results from 120 mg HDA notably showed the dominant fibrous structure of CuNWs, and the average diameter and length for CuNWs were 66.4 ± 16.1 nm and 32.4 ± 11.8 µm, respectively. The estimated aspect ratio (L/D, length-to-diameter) was 488 ± 215. This high aspect ratio usually implement a strong anisotropic behavior in the electrical and magnetic properties might possibly open up prospects for fundamental studies of nanoelectronics and phase change memories [21,22].

The dependence of the measured average diameter of CuNWs on the content of dispersion agent HDA was summarized in Table 1. The decreasing average diameter of harvested CuNWs revealed a similar trend with the UV-Vis maximum absorption peak lowering. However, the average diameter of CuNWs showed no statistical difference (*p* > 0.05) among the HDA dosages of 105, 102, and 135 mg. The growth behavior of Cu nanocrystal is governed by the ratio between the directional growth rate of each facet [23]. M. Jin et al. reported the usage of capping agent of poly(vinyl pyrrolidone) (PVP) selectively adsorb on the (100) facets of Ag, Au, Pd, or Pt) to enhance the harvest of nanocubes, nanobars, or penta-twinned nanowires [24]. B. Jia et al. reported the CuNWs preparation from a capping agent of tetradecylamine (TDA) under various contents. They demonstrated a capping agent’s versatility in controlling a metal nanoparticle’s shape [25].

Before industrial nanowires applications, excellent dispersion is the crucial step for nanocomposite reinforcement. The nonpolar nanowires usually aggregate due to the hydrophobic effect caused by excluding nonpolar moieties from an aqueous environment [26]. Distribution is known as how the particles fill the space, whereas dispersion is whether these particles are agglomerated or not. With a good distribution, each particle is segregated as far as possible from its nearest neighbor, so the particles are homogeneously filled in space. To improve the dispersion, the surfactant of PEG-7 was chosen and investigated the dispersion behavior of CuNWs suspension. Figure 4 revealed the effect of surfactant PEG-7, on the homogeneity of CuNWs suspensions. Unlike Figure 4a showing the aggregation of CuNWs, Figure 4b demonstrate a good dispersion of CuNWs with PEG-7 in an optical micrograph. To further investigate phase behavior for CuNWs/PEG-7/water system, a processing map was established and classified into four sections of (a)~(d) according to their status of dispersion and distribution (Figure 5). The addition of PEG-7 tended to enhance the homogeneity of CuNWs suspension. Unlike region A, which revealed an aggregate of CuNWs with poor dispersion, region D represented a condition of good dispersion and distribution. The concentration of CuNWs might play a key role in dispersion. It showed good homogeneity at the CuNWs concentration as low as 0.1%, while the surfactant PEG-7 tended to improve the 0.5% CuNWs suspensions distribution. 

The i-PP/CuNWs composites were prepared by solution blending under solvent of p-xylene due to its good solubility of i-PP polymers. Additional melt mixing was conducted to remove the pores from solvent evaporation and to prepare uniform samples with an excellent CuNWs dispersion for further electrical volume resistivity measurement. The thermal analysis, crystalline morphology, crystalline microstructures, and of CuNWs composites were further investigated by DSC, POM polarized optical microscopy (POM), XRD, and SEM, respectively. 

### 3.2. Transcrystallization of i-PP/CuNWs Composites

Isotactic PP (iPP) is a semicrystalline polymer with relatively low crystallization speed under homogeneous nucleation, leading to the development of micron-sized crystals and poor transparency [27]. Figure 6 shows the optical micrographs of i-PP/CuNWs composite first heating to 200 °C, then cooling to room temperature. Unlike the micrographs revealed no birefringence at 200 °C and 123 °C (Figure 6a,b) at the molten state, the specimen showed sign of spherulites under polarized optical microscopy when cooling 117 °C and 111 °C (Figure 6c,d). The crystallization temperature of the i-PP/CuNWs composite is higher than that of pure i-PP, indicating the possible transcrystallization. Typically, transcrystallization is recognized as a nucleation-controlled process occurring under quiescent conditions when a semicrystalline polymer is in contact with other materials [28]. G. Pompe et al. claimed a high heterogeneous nucleation ability of the surface the closely spaced nuclei force growth in one direction for glass-fiber/polypropylene composites [29]. B. Wang et al. [30] reported the POM photographs about transcrystallization of the iPP/Bacterial cellulose (BC) composites at 138 °C. The crystallization firstly starts at the interfaces of the bacterial cellulose, then extends to the iPP layers, and eventually forms the transcrystals. Figure 6e was morphology for molten i-PP/CuNWs composite cooling down to room temperature. This unique morphology evolution might possibly be attributed to the competition of crystallization and phase separation or the crystallization-induced phase separation. 

### 3.3. DSC Thermal Analysis for i-PP/CuNWs Composites

Figure 7 show the DSC curves for various i-PP/CuNWs composites. The DSC heating curve (Figure 6a) showed the endothermic peak at 162.5 °C for the pure i-PP, but the melting temperature (*Tm*) decreased with increasing the CuNWs content for i-PP/CuNWs composites. The low melting peak down to 144.2 °C was observed in the case of CuNWs content at 1.25 vol%. On the other hand, the crystallizing temperature (*Tc*) of i-PP/ CuNWs composites increased up to approximately 5 °C with the addition of 1.25 vol% of CuNWs, implying the character of CuNWs as a nucleating agent or enhancing the thermal conductivity. 

The summary of thermal analysis for various i-PP/CuNWs composites is listed in Table 2. The melting point decreased and the crystallization temperature increased with increasing the CuNWs content. However, the melting enthalpy first decreased with increased CuNWs content till 0.75 vol%, then slightly increased up to 1.25 vol%. A similar trend was observed, even normalized melting enthalpy with the weight of CuNWs. 

From the thermodynamic point of view, the melting point depression of a crystalline polymer due to the presence of a miscible diluent [31]. For an i-PP/CuNWs composite, the i-PP is crystallizable. However, the CuNWs might not be able to serve as a miscible diluent due to their metal nature and high melting point. There must be some other reasons for this peculiar melting point depression phenomenon. The theory of the molecular weight (MW) dependence on equilibrium melting temperature was proposed by Flory and Vrij due to the effect of entropy of chain ends [32]. Additional tests are needed to elucidate the current results.

To further verify the possible mechanism of melting point depression, the DSC spectra for i-PP/CuNWs composites with 1.25 vol% CuNWs under various mixing periods on a hot plate were recorded (Figure 8). The melting temperate (*Tm*) for solution cast i-PP and i-PP/CuNWs were around 164 °C, while the *Tm* decreased to 144 °C and 109 °C after 1 min and 5 min melt mixing, respectively. Under such circumstances, the thermal degradation of i-PP/CuNWs composites during the melt mixing could be one of the possible routes. 

### 3.4. XRD Analysis for i-PP/CuNWs Composites

To elucidate the microstructure of CuNWs, i-PP, and i-PP-based composites with 0.1 vol% and 1.25 vol% content of CuNWs, the XRD patterns were carried out and shown in Figure 9. All the characteristic diffraction peaks for i-PP with 2θ angles of 14.2°, 17°, and 18.8° were attributed to the pure α-form crystal planes of (110), (040), and (130) [33]. Commercial iPP crystallizes, prepared from the traditional heterogeneous Ziegler–Natta catalytic system, usually reveal the stable α form under the most common conditions [34]. All the diffraction peaks with 2θ values of 43.4°, 50.4°, and 74.1° correspond to the crystal planes of (111), (200), and (220) (JCPDS # 04-0836) for pure CuNWs. The capping agent HDA selectively adsorbed onto the (100) facets cubes is known to have a preferred orientation with (100) planes, exhibiting a (111) peak with a higher intensity than (200). On the other hand, the 2θ diffraction peaks of 36.4° and 61.5°, corresponding to (111) and (220) for crystal planes of Cu_2_O, indicate the possible oxidization occurring in the melt blending of i-PP/CuNWs composites (JCPDS # 99-0041) [35]. Compared with other noble metals, Cu is relatively active and easy to react with oxygen to form a thermodynamically stable oxide layer with a deterioration of its electrical properties. The major limitation of the CuNWs preparation is their ease of oxidation to less conductivity form of CuO or Cu_2_O without providing an inert environment [36]. An additional thermal reduction process could solve this undesirable situation in a hydrogen atmosphere [37]. The transformation of pure CuNWs into detectable Cu_2_O after melt mixing is likely to happen. It might play a role in thermal degradation for i-PP/CuNWs composites, evidenced by the melting point depression. 

### 3.5. Electrical Volume Resistivity Measurement for i-PP/CuNWs Composites

The measured volume resistivity was 5.44 × 10^14^ Ω-cm for i-PP. From previous results, the estimated aspect ratio (L/D, length-to-diameter) was 488 ± 215. The volume electrical resistivity of i-PP containing metal nanowires decreases by about eight orders of magnitude for CuNWs (Figure 10). At concentrations of 0.25–0.5 vol%, the resistivity of the nanocomposites depends primarily upon the connectivity of conductive CuNWs. Next, Figure 11 shows the electrical percolation threshold volume fraction Vc of 0.00237 and critical exponent of 2.96 after fitting with the power law equation [38,39,40]. B. Genaro reported the electrical percolation threshold for silver (L/D~200) and copper ((L/D~400)) nanowires in polystyrene composites were 0.50–0.75 vol% and 0.25–0.75 vol%, respectively.

Power law equation: ρ = ρ_o_ (V − Vc)^−t^
ρ = Composite electrical resistivity
ρ_o_ = Scaling factor
V = Volume fraction of filler
Vc = Electrical percolation threshold volume fraction
t = Critical exponent revealing the lattice dimensionality

It is always desirable to reduce the electrical percolation threshold to reduce the dosage and cost of conductive nano-fillers for composite applications. Based on the concept of percolation threshold, the connective network becomes conductive when the conductive nanofillers’ critical concentration is reached. Moreover, we should re-examine the processing criteria for an excellent conductive nanocomposite. Unlike the traditional composites perusing good dispersion and distribution of fillers to reach maximum reinforcement, a conductive composite might need a preferential allotment of well-dispersed fillers to form a connective network with good electrical conductivity. Poor dispersion of the fillers prohibits network formation, while a perfect distribution of well-dispersed fillers increases the distance between the neighboring fillers, thus restricting network formation [41]. Furthermore, the electrical percolation threshold investigation also provides a valuable tool for designing a thermally conductive but electrically insulated polymer composite containing 3 vol% CuNW that reaches high thermal conductivity of 0.50–1.04 W/mK [42].

## 4. Conclusions

The characterizations determined the successful synthesis of the CuNWs and its i-PP/CuNWs composites through UV-Vis spectra and SEM. The DSC characterization of the thermal properties of i-PP/CuNWs composites. The high melting point depression of i-PP/CuNWs after long melt mixing treatment and the XRD diffraction peak of Cu_2_O imply a possible thermal degradation of i-PP. A low electrical percolation threshold value of 0.237 vol% and high critical exponent t of 2.96 for i-PP/CuNWs composites were obtained. The volume resistivity for i-PP/CuNWs composite was 1.57 × 10^7^ Ω-cm at 1 vol% of CuNWs, indicating a potential candidate for future conductive materials. The phase behavior window of nanocomposites related with the dispersion and distribution of nanofillers might offer a useful tool for processing.

## Figures and Tables

**Figure 1 polymers-14-03369-f001:**
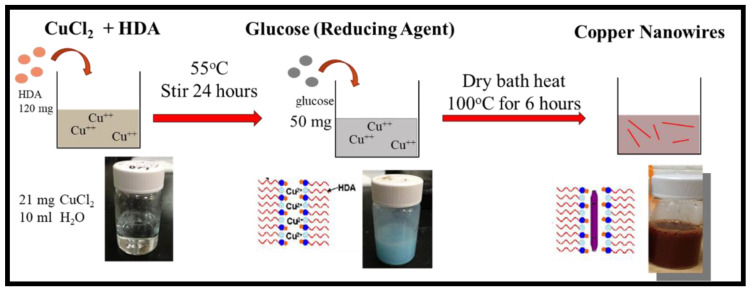
The preparation procedure of copper nanowires (CuNWs).

**Figure 2 polymers-14-03369-f002:**
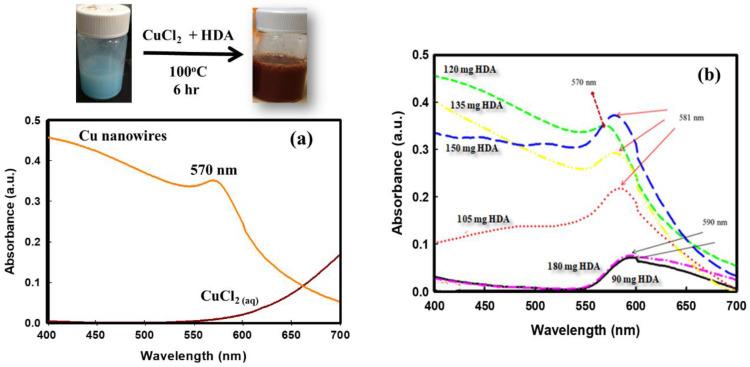
UV-Vis spectra and photographs of the samples of the CuNWs solutions. (**a**) UV-Vis spectra for both CaCl_2_ and CuNWs solution. (**b**) UV-Vis spectra of CuNWs solutions under various HAD contents.

**Figure 3 polymers-14-03369-f003:**
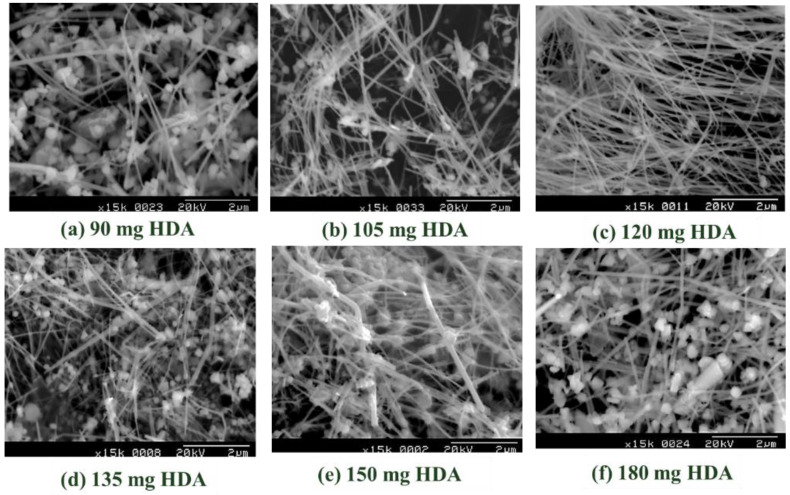
The SEM results of capping agent HAD content dependence on the morphology of copper nanowires.

**Figure 4 polymers-14-03369-f004:**
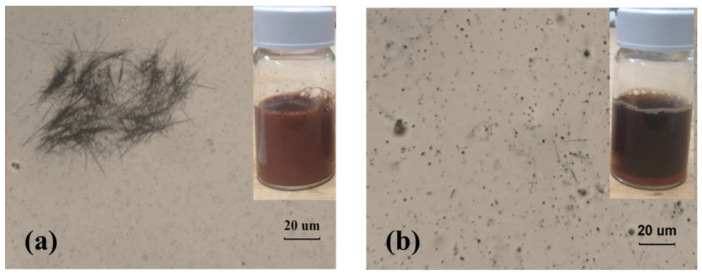
The homogeneity of CuNWs in suspension (**a**) without dispersion agent, (**b**) with 1 wt% PEG-7.

**Figure 5 polymers-14-03369-f005:**
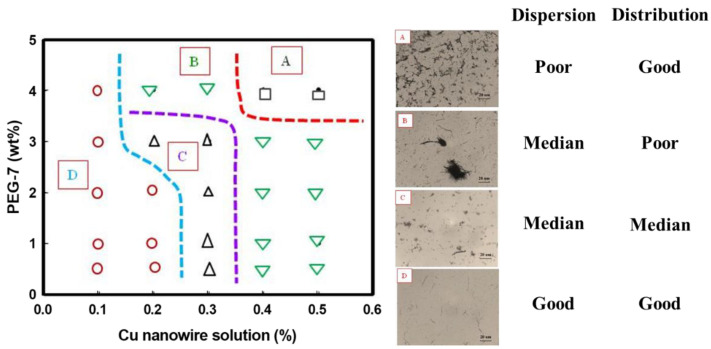
The phase behavior window for CuNWs/PEG-7/Water.

**Figure 6 polymers-14-03369-f006:**
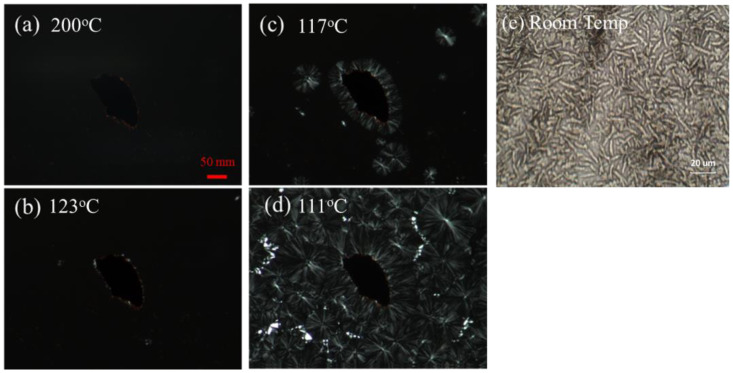
The optical micrographs of transcrystallization in solution-cast i-PP/CuNWs composites during cooling from (**a**) 200 °C to (**b**) 123 °C, (**c**) 117 °C, (**d**) 111 °C, and (**e**) room temperature.

**Figure 7 polymers-14-03369-f007:**
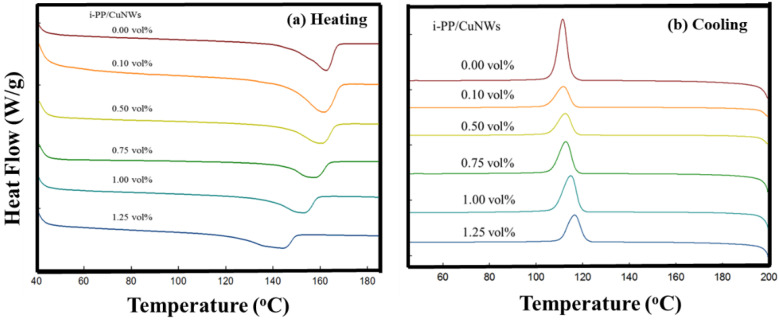
The DSC spectra of solution-cast i-PP/CuNWs composites.

**Figure 8 polymers-14-03369-f008:**
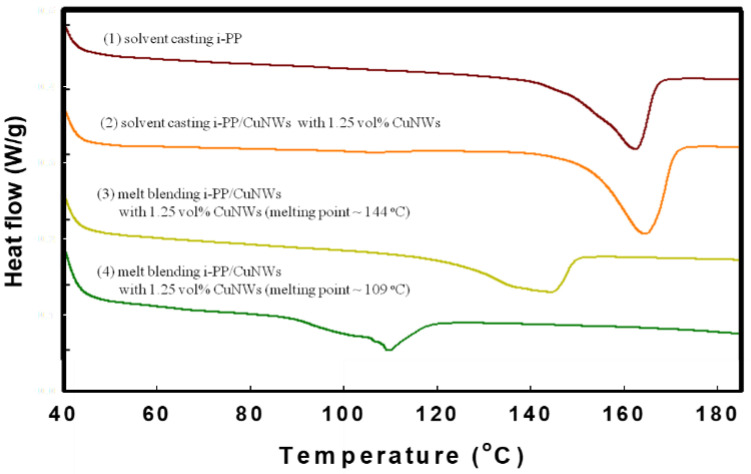
The DSC thermographs for (1) solution casting i-PP and (2) solution blending i-PP/CuNWs (1.25 vol%) composites, (3) 1 min melt blending time, and (4) 5 min melt blending time.

**Figure 9 polymers-14-03369-f009:**
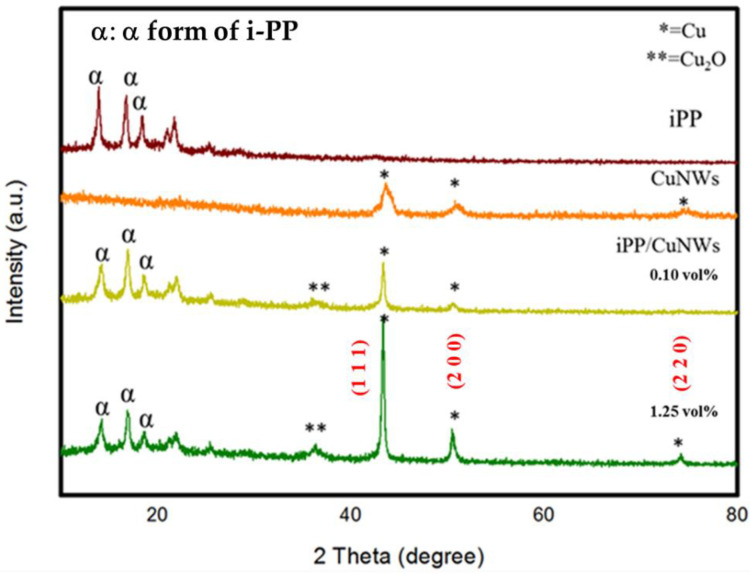
XRD pattern of various i-PP/CuNWs compositions.

**Figure 10 polymers-14-03369-f010:**
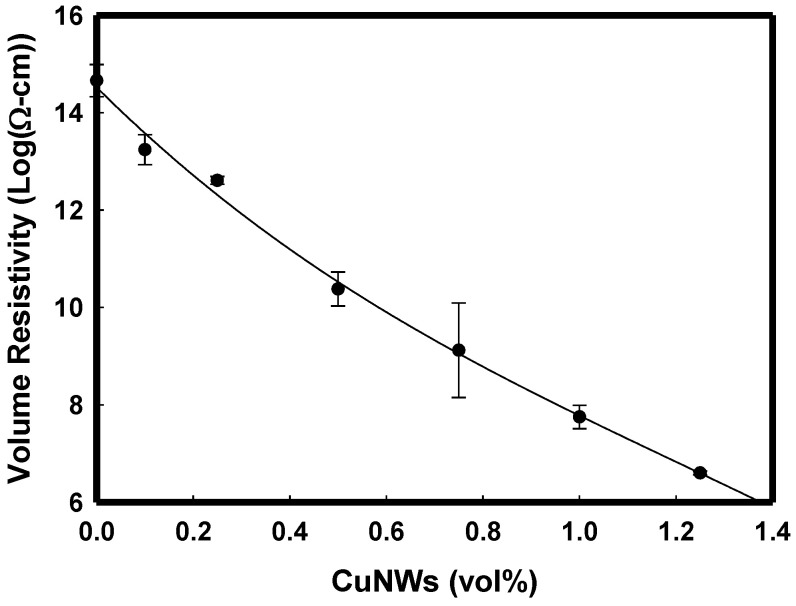
The CuNWs content is dependence on the volume resistivity for various i-PP/CuNWs composites.

**Figure 11 polymers-14-03369-f011:**
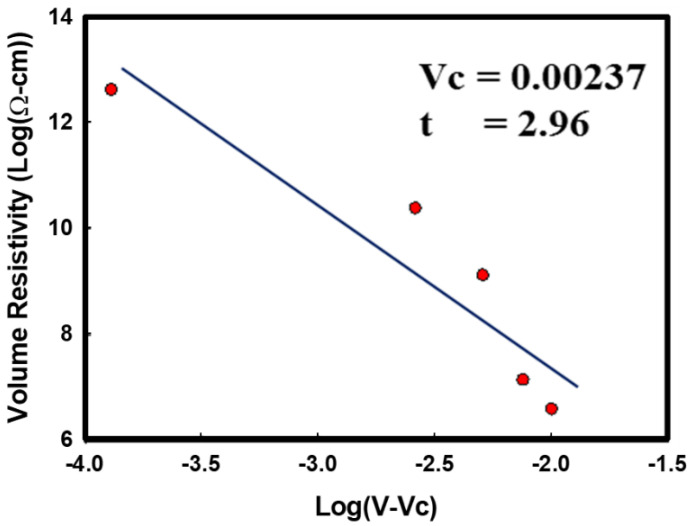
The electrical percolation threshold of CuNWs for i-PP/CuNWs composites.

**Table 1 polymers-14-03369-t001:** The dependence of average diameter of CuNWs with dispersion agent HDA content. Diameter of CuNWs (N = 20). Values are the mean ± standard deviation. Mean values followed by the same superscript letter do not significantly differ (*p* > 0.05) according to post-hoc test.

HAD Content (mg)	Absorbance Peak (nm)	Average Dia. CuNWs (nm)
90	581	110 ± 22 ^a^
105	570	85 ± 20 ^b,d,e^
120	570	66 ± 16 ^c,d^
135	581	80 ± 18 ^d,e^
150	581	92 ± 21 ^e^
180	590	115 ± 18 ^a^

**Table 2 polymers-14-03369-t002:** The summary of thermal analysis for solution-cast i-PP/CuNWs composites.

CuNWs (vol%)	*Tm* (°C)	∆H (J/g)	Normalized ∆H (J/g)	*Tc* (°C)
0.00	162.5	64.8	64.8	111.5
0.10	161.1	63.2	63.8	111.7
0.50	160.7	54.4	57.1	112.6
0.75	157.7	46.8	50.3	112.8
1.00	153.1	49.4	54.4	114.9
1.25	144.2	50.1	56.4	116.6

## Data Availability

Not applicable.

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
