# Peer review of "The Processing and Electrical Properties of Isotactic Polypropylene/Copper Nanowire Composites"

_polymers, 2022, doi:10.3390/polym14163369_

Round 1

Reviewer 1 Report

The authors performed a systematic study of the structure-electrical properties relationship of Isotactic Polypropyl-2 ene/Copper Nanowire Composites. The manuscript is well written and the results reported by the authors are intriguing.

I have following two comments:

 1.       Did the authors perform EDX mapping of the Nanowires in addition to the SEM scanning?

2.       It is observed that the Nanowires show a strong anisotropic behavior in the electrical and magnetic properties (https://doi.org/10.1038/s41598-020-65805-4, https://doi.org/10.1021/nl080524d ) did the authors observe such an anisotropic behavior in the electrical properties?

Reviewer 2 Report

This study investigates the thermal and electrical properties of isotactic polypropylene/copper nanowires (i-PP/CuNWs). The CuNWs were harvested by chemical reduction of CuCl2 using a reducing agent of glucose, capping agent of hexadecylamine (HDA), and surfactant of PEG-7 glyceryl cocoate. Their morphology, light absorbance, and solution homogeneity were investigated by SEM, UV-visible spectrophotometry, and optical microscopy. The research design is appropriate and complete, the results are clearly presented and the conclusions are supported by the results.

(1) For part 3.2, it is suggested cite the references 10.3390/polym11030508.

(2)Line 275. 1.25 vol%. what is the mass percentage of the CuNWs?

(3) The Tm and Tc should Italics and angle marks.

(4) Line 293-299 need more discussion.

(5) The Conclusions part need more discussion.
